# A Narrative Inquiry into the Practices of Healthcare Workers’ Wellness Program: The SEED Experience in New South Wales, Australia

**DOI:** 10.3390/ijerph192013204

**Published:** 2022-10-13

**Authors:** Katarzyna Olcoń, Julaine Allan, Mim Fox, Ruth Everingham, Padmini Pai, Lynne Keevers, Maria Mackay, Chris Degeling, Sue-Anne Cutmore, Summer Finlay, Kristine Falzon

**Affiliations:** 1School of Health and Society, The University of Wollongong, Northfields Ave, Wollongong, NSW 2522, Australia; 2Rural Health Research Institute, Charles Sturt University, Leeds Pd, Orange, NSW 2800, Australia; 3Illawarra Shoalhaven Local Health District, 67-71 King Street, Warrawong, NSW 2502, Australia; 4School of Nursing, The University of Wollongong, Northfields Ave, Wollongong, NSW 2522, Australia; 5Waminda South Coast Women’s Health and Welfare Aboriginal Corporation, 122 Kinghorne St, Nowra, NSW 2541, Australia

**Keywords:** healthcare workers, workplace wellness, mental health and wellbeing, recovery, resilience, Australian bushfires, COVID-19, burnout, occupational trauma

## Abstract

The 2019–2020 Australian bushfires followed by the COVID-19 pandemic brought the significant mental health implications of working in healthcare to the fore. The importance of appropriate support services to ensure the resilience and recovery of healthcare workers has been highlighted. In response to healthcare staff experiences during the bushfires, the SEED Wellness Program was created in 2020 in the Illawarra Shoalhaven Local Health District in New South Wales, Australia. SEED used a participant-led design to engage healthcare staff in workplace-based restorative activities. Guided by practice theory, this study aimed to identify and describe SEED wellness practices that supported healthcare staff. Thirty-three healthcare workers participated in focus groups or individual interviews between June 2021 and March 2022. The analysis involved inductive thematic individual and collective exploration of SEED practices, including co-analysis with participants. Eight core practices that supported participants’ wellbeing were identified, including responsive and compassionate leading, engaging staff at every stage of the recovery process, creating a sense of connection with others, and collective caring. The study found that workplace wellness initiatives are optimised when they are place-based and grounded in local knowledge, needs, and resources incorporating a collective and supportive team approach. Moreover, to ensure engagement in, and sustainability of these initiatives, both bottom-up and top-down commitment is required.

## 1. Introduction

The impacts of the 2019–2020 Australian ‘black summer’ bushfires followed by the COVID-19 pandemic illustrate the significant mental health implications of crises [1], and the importance of immediate and appropriate support service provision to ensure the recovery of individuals and communities during and following these events. According to the World Health Organisation [2], almost all people affected by disaster will experience some level of emotional or psychological distress. Previous studies on the impacts of bushfires in Australia revealed that rates of mental health problems, such as depression and post-traumatic stress disorder (PTSD), remained higher than national levels five years later [3]. In Australia, the COVID-19 pandemic has created unique circumstances by not only interrupting the early stages of post-bushfire recovery but also exacerbating psychological distress in the affected regions [1].

Healthcare workers (HCW) require particular attention in the discussion about post-crisis resilience and recovery. HCW are in an industry prone to stress and burnout because of their exposure to human suffering, the emotional demands of caring for people, low levels of social support at work, and the pressures of working relationships with patients, families, and employers [4,5]. In New South Wales (NSW), HCW make up the largest group of mental-stress related claims on workplace insurance [6]. Across the world, HCW have carried the burden of responding to COVID-19 morbidity and mortality and associated impacts on health care system capacity. Working under new pressures and constantly changing environments, taking on additional shifts, risking their own health and lives, and experiencing ongoing psychological and emotional strain are ubiquitous [7,8]. In many regions in Australia, HCW have carried the additional burden of having to respond to the multiple crises while dealing with any personal adversity and trauma they might have been experiencing as a result of the bushfires and pandemic. Many studies have quantified the psychological impact of natural disasters [9,10] and the pandemic [11,12] on HCW and identified the need for workplace interventions to mitigate those impacts. Hobfoll et al. [13] identified five essential elements of immediate and mid-term mass trauma intervention: (1) a sense of safety, (2) calming, (3) a sense of self- and community efficacy, (4) connectedness, and (5) hope. These principles have been described as relevant for managing the stressors during the pandemic [14]. However, few workplace programs to enhance HCW resilience and recovery have been identified or evaluated [15,16,17,18], and no studies to date have examined health workplace responses to a co-occurring natural disaster and pandemic. 

In this article, we document the practices of a hospital staff wellness program named SEED (Stability, Encompassing, Endurance and Direction) developed and implemented to address bushfire and pandemic related trauma in the Illawarra Shoalhaven Local Health District (ISLHD), NSW, Australia. ISLHD delivers health services to mostly rural and regional communities that were heavily impacted by the 2019–2020 bushfires. Eighty percent of the southern part of the district was burnt, with hundreds of homes destroyed or damaged [19]. The fires lasted for 74 days until extinguished in February 2020 by heavy, and then ultimately and ironically, flooding rains. Within a month (March 2020), COVID-19 measures and restrictions were introduced in Australia, and HCW in ISLHD then faced the unprecedented task of responding to the pandemic. SEED used a strengths-based approach to ascertain the needs of staff and implement staff-led wellbeing initiatives that build resilience and aid recovery processes. Some of the SEED initiatives that were implemented across different ISLHD sites include: (1) Coffee Buddies—a planned coffee break with a nominated staff member as a form of collective care for each other, (2) Quiet Room—a space inside the hospital for staff to take a moment each day for a quiet reflection, (3) 24/7 Wellness—weekly wellness sessions in the workplace to promote staff wellbeing, (4) (S)Crap book—a communal journal to share and reflect on stressful stories occurring during the pandemic (for details on the SEED model, history, and activities, see [20,21]). 

### Wellness and Resilience of Healthcare Workforce

The resilience of the global healthcare workforce has been tested by COVID-19. In the context of workforce, resilience has been defined as “the ability to adapt and reorganise to maintain core functions and activities” ([22], p. 3). Studies on healthcare staff across the world demonstrate increased stress and anxiety [8,23], emotional exhaustion [24], compassion fatigue and burnout [25,26], a higher intention to leave the organisation [27], and an overall reduction in the quality of care delivered due to COVID-19 [28]. These studies unanimously call for workplace interventions to address the negative impact of COVID-19 on healthcare staff and to prevent the onset of burnout and support staff retention [23,24]. 

Prior to the pandemic, workplace wellness initiatives typically focused on individuals addressing their own lifestyle risk factors, such as diet and exercise, outside of the workplace [29]. A few studies during COVID-19 examined the feasibility, acceptability, and effectiveness of interventions aimed at wellness and resilience of healthcare staff [15,16,30]. These studies found high satisfaction amongst participating staff and increased self-reported resilience [30]. Based on a systematic review of interventions to support the resilience and mental health of frontline healthcare staff during and after a disease outbreak, epidemic or pandemic, Pollock and colleagues [31] concluded that the most important facilitators of program implementation were adaptability to local needs; effective communication, both formally and socially; and having positive, safe and supportive learning environments for frontline workers [31]. Conversely, some of the barriers were lack of awareness, by both staff and organisations, of the need to support staff mental health, as well as lack of equipment, staff time, or skills needed for an intervention [31]. Recovery from mental health problems following a disaster (such as a pandemic) requires community-level interventions that acknowledge shared experience [32]. However, healthcare organisations still have little guidance on how they can support staff experiencing workplace crises.

To address the gap in research on workplace wellness interventions in healthcare settings, this study explored the experiences of staff participating in a workplace wellness program (SEED). Specifically, this study aimed to (1) identify and describe the core SEED practices that supported HCW to recover following bushfires, floods, and COVID-19, and (2) document the benefits and challenges of the staff wellness program described by study participants.

## 2. Methodology

This study was guided by practice theory, where the primary unit of analysis is practice, described by Schatzki [33] as the complex interactions of sayings, doings, and relatings between people, other beings, and material artefacts. Schatzki [33] argues that practices provide the basic structure of social life in social sites. Accordingly, healthcare institutions, such as ISLHD, are social sites comprising practices and material-economic, socio-political, and cultural-discursive arrangements that persist and frame practice possibilities [34]. Healthcare settings are turning to practice theorisations in recognition of the need for practice-based evidence from everyday clinical and organisational settings to complement the evidence obtained from formal randomised controlled trials [35]. Therefore, a practice-based approach for this study focuses on the situated and embodied ways that staff at local hospitals acted to enhance their mental health, wellbeing, healing, and recovery.

This practice-based study utilizes participatory narrative inquiry (PNI) [36] to learn with and from the partner organisation about the wellness practices they employed during the disasters and, later, COVID-19. PNI is how the practices were defined and described. PNI is an approach that is inclusive of multiple stakeholders and involves individual and collective exploration of, and critical reflection on, practices and interventions to shape future collective endeavours. 

### 2.1. Study Setting

The study reported here is part of a larger research project which documented and compared the crisis responses of two healthcare organisations: a local health district (ISLHD) and an Aboriginal Community Controlled Health Organisation. This article focuses on the ISLHD findings only. The ISLHD is the largest provider of health services in the Illawarra and Shoalhaven regions, servicing 400,000 residents [37]. Publicly funded, it spans 250 km of the southern coastal strip of NSW and operates eight hospitals in the region [38]. This study focused on five hospital and administration sites where the SEED program was delivered. SEED was created from the urgent need to address the psychological impact of bushfires on affected HCW. The development of SEED was guided by a strengths-based approach [39] and post traumatic growth (PTG) theory [40,41], which aimed to ascertain the needs of staff and implement staff-led wellbeing initiatives that build resilience and aid recovery processes. Support for the SEED wellness program at the initial site—the Milton Ulladulla Hospital (MUH)—as evidenced in the results of an internal evaluation, resulted in ISLHD responding to the impact of COVID-19 by adapting and implementing the SEED interventions in other hospitals in the district. This study investigated the practices that led to SEED’s success at MUH and its subsequent adaption and implementation at other sites.

### 2.2. Recruitment and Sample Characteristics

The study used purposive sampling to recruit participants who (1) were 18 or older, (2) were hospital staff within ISLHD, and 3) had participated in the SEED program. The recruitment letter was circulated to ISLHD staff via email contact details of SEED participants obtained from the SEED program lead. Forty-four HCW from ISLHD agreed to participate; however, due to COVID-19 restrictions on mobility, the final number of participants was 33. Table 1 provides the characteristics of the sample, which is representative of the ISLHD healthcare workforce [42]. 

### 2.3. Data Collection

Data collection took place between May 2021 and March 2022. Two focus groups were conducted with MUH staff (*n* = 10), and the remaining 23 participants took part in semi-structured individual or group interviews. Data collection was conducted by one or two members of the research team. The focus group and interview guides were developed collectively by the whole research team to explore staff experiences with the bushfires and COVID-19 and their perceptions of the SEED program (see Appendix A). 

The focus groups were conducted face-to-face on the hospital site and ranged from 110 to 120 min in length (average of 115 min). Subsequently, due to COVID-19 lockdown restrictions, the interviews were conducted via Zoom, and ranged from 28 to 110 min in length (average of 56 min). All data collection sessions were recorded and professionally transcribed. The researchers took fieldnotes reflecting on the process and the outcome of each focus group and interview. These were discussed in weekly team meetings, which allowed for monitoring of the quality of the interviews and making needed improvements such as rephrasing questions or using alternative prompts [43]. Data collection also involved ethnographic observations of an all-day SEED event hosted at one of the hospitals (pre-COVID restrictions). The first author took extensive fieldnotes documenting the day’s events, group discussions, and interactions [44]. Finally, notes and photos capturing the SEED sociomaterial arrangements and artefacts, such as *The Quiet Room,* were taken.

### 2.4. Data Analysis

The analysis involved the research team using inductive individual and collective exploration of SEED practices described in the interviews and focus groups. Specifically, the data were analysed using an inductive thematic approach [45] and organised and managed through the NVivo software. The authors utilised Braun et al.’s [45] six stage thematic analysis approach which involved familiarisation with the data by reading and re-reading the interview transcripts and utilising NVivo to systematically generate codes and manage the data. Five members of the research team conducted preliminary analysis of twenty-four transcripts, each identifying and defining codes. The questions asked of each transcript were: How is SEED described? What are its practices?What do people say about their own or others’ wellness or un-wellness and ways that SEED influenced that?How are the absence or presence of resources described?What relationships between people are discussed?How do people do things differently because of SEED?

Seventy initial codes were created. The research team reviewed each separate analysis in NVivo identifying similarities and differences in coding and code definitions to remove duplicate codes and cluster the remaining 52 codes into themes (see Appendix A). Themes were then constructed, revised, and defined during weekly team meetings to capture “a meaningful pattern across the dataset” ([45], p. 855) and to gain a strong understanding of how each theme relates to another to define SEED praxis. Two interpretive focus groups [46] (*n* = 16) were then conducted to present preliminary findings to study participants and seek their interpretation of the themes. Questions asked included, “what do you think is the most important finding”, “is there anything missing” and “anything you disagree with”? The groups were recorded, transcribed, and incorporated into the analysis. The co-analysis ensured that practices emerging from the research team’s analysis of the transcripts were consistent with the participants’ lived experiences of the SEED program [47]. Co-generation and interpretation of data is a critical element of PNI and ensures credibility and trustworthiness of the study findings [48]. Finally, we triangulated the data by checking for consistency across the multiple data sources [43]. 

## 3. Results

As a culmination of the individual and collective data analysis processes, eight practices of the SEED Wellness Program were identified: (1) responsive and compassionate leading, (2) co-designing wellness activities with staff, (3) listening to understand, (4) creating a safe and healing space, (5) connecting with others, (6) collective caring, (7) diversifying and localising wellness activities, and (8) striving for sustainability. The boundaries made between practices in analysis were heuristic devices to make them distinguishable. Nevertheless, these practices are interwoven and build on each other. The description of each practice includes the key elements of the practice and examples of the benefits and implementation challenges experienced by participants. To protect the confidentiality of the research participants, they will be referred to as “staff” (S) and identified by their assigned number when quoted, e.g., S16 

### 3.1. Responsive and Compassionate Leading

This practice refers to the immediate recognition and response of the leaders to the bushfire crisis and its impact on staff wellness. Recovery-oriented activities to respond to staff needs were initiated by the leaders, for example, by sending an experienced social worker to MUH immediately upon recognition of the devastation the bushfires were causing to communities and the impact that was having on hospital staff. As one leader explained: 

“*We needed to come up with a plan as to, how are we going to cope going forward. And so [Leader] made some suggestions about connecting with the crew and within a very short period of time, we had run sessions, understood what people were actually wanting”.*(S01)

Staff who experienced this response noted the positive and significant impact the immediate organisational response to disaster had, 

“*…the most important thing was that thought was given and it was instantaneously accommodated, which brought the faith and trust and my belief in ‘I’m validated and heard and that, wow, someone’s listening to concerns but the solution to it is happening quite quickly’”.*(S16)

It was critical to SEED implementation at all sites that leaders demonstrated ongoing support for staff wellness and their own wellness following initial activities by both supporting the program practically with resources and role modelling by participating in the wellness initiatives themselves: “*the leadership has to support it and role model it*” (S30). The benefit of having SEED engaged leaders was that they considered staff wellness as core business by allocating time for wellness activities during work hours such as the all-day SEED event observed in this study. Two leaders attended and actively participated in the day’s wellness activities and discussions. This facilitated staff participation through both explicit and tacit permission:

“*It was giving people permission and I think I was one of those people, as a leader that just went, ‘We can do this. It’s okay to do this. You can do this [participate in SEED] because this is what we need. This is collectively what we need, and you have my support to do it’*”.(S21)

The challenges around the leadership practice were two-fold. First, if a site leader did not support SEED, it was virtually impossible to implement the practices because they were workplace and team based so necessarily occurred during work time: “*You’ve actually got to deal with the leadership issue because to impose something on an area where the leader doesn’t believe in it, it’s not going to work*” (S33). The second challenge was the impact of SEED leadership on leaders themselves. The responsibility to make resources, including time for SEED during work, and to demonstrate participation as a way of facilitating others to join was experienced as a burden by some in spite of their general support for the program:“*You need that leadership and that one person to keep putting it at the front of their attention and it’s been the one thing that’s actually burnt me out this year, even amongst COVID and everything else*”.(S24)

### 3.2. Co-Designing Wellness Activities with Staff

The SEED Wellness Program was co-designed with the staff. Consultations with staff in all roles took place at each site from the very beginning to ensure the wellness program reflected staff needs. It also aimed to create a sense of ownership in staff of their wellness-oriented activities. The process was entirely participatory, inclusive, and collaborative. Whereas the co-design process at the sites affected by the bushfires and floods was initially recovery and resilience oriented, it was adapted with time to respond to any staff wellness needs at a given site. As explained by one participant, “*what SEED does differently is that it asks the participants what they need and what they want, and then the strategies are built around that*” (S30). The benefits of inclusive and genuine staff co-design were emphasised by another person: “*I think that’s the biggest learning of SEED, it really has to come from what is important to the staff*” (S19). 

The two challenges of staff co-design were including as many people as possible and maintaining the practice. As SEED constantly evolved in each site and people came and went from activities, consultation had to continue to ensure that wellness activities were responsive to staff needs and preferences over time: “*It’s listening to the staff about what they need at that time, and continually customising wellness to meet the team’s needs*” (S30). Given the high staff turnover, there was a need for constant reiterating. In addition, although SEED co-design was inclusive of all staff roles in a site, not everyone wanted to participate. For some staff it was noted as “*not their thing*” (S21), but for others workplace change was not welcomed:

“*There is entrenched concern or cynicism from the staff, certainly when new programs come out it’s always with suspicion, ‘What are they trying to make me change, what’s this in a consequence of, what am I going to be measured on?’”.*(S25)

### 3.3. Listening to Understand

Another core practice of wellness described by the staff was deep listening. Staff were taught how to hold space for one another and deeply listen to what the other person is saying with the intent to understand rather than to fix. This was especially important as staff were dealing with some traumatic experiences such as losing their houses in the fires. This is how one participant described the practice of holding space and listening to their colleagues:

“*…the leader [SEED facilitator] showed us the way, gave us some tools to hold the space, which we could use, especially for people that lost their house. Because that would come up and you didn’t know what to say, but boy, sitting there listening, holding that space, sort of did … gave them a fair bit of relief, I suppose, which was rewarding once you knew that you were actually doing it. So being shown how to hold the space was one of the biggest things I got out of it”.*(S06)

The benefits of listening to understand were for those listening as well as those being heard. This practice took the pressure off individuals to fix another’s problem or provide a solution but gave them direction about how to respond to post-disaster distress: “*I think I have become a better listener and, just ways of trying to encourage people to open up and share their experiences and let themselves, let them be vulnerable without being judged*” (S14). Through this practice participants learned how to resist their tendency to give answers: “*they just let people be sad, and held the space by keeping their quietness, nobody tried to solve or give an answer*” (S12). The challenge of listening to understand was around staff learning a new and unfamiliar practice that caused initial discomfort: “*We went and held space. We were really uncomfortable, but we were taught to sit through these uncomfortable feelings, and it made us stronger with them*” (S03). 

### 3.4. Creating a Safe and Healing Space

Opportunities and activities were created that allowed staff to share their stories of bushfire experiences and recovery and the subsequent experiences with COVID-19. This practice was guided by the intention of creating a safe space where staff could begin their recovery from the crises they experienced. Story sharing was integrated into various SEED activities, including the *Wellness Wednesdays* and *Wellness Warriors*. This is how one participant described this practice: “*And then [leader] had arranged like an open discussion group. I think there was about six or seven staff and they were just sharing their stories. So, it wasn’t any great structure, it was listening to the different stories*” (S19). Story sharing allowed staff to be authentic, emotional, and vulnerable, and feel heard without being judged. One of the staff described their experience in this shared safe space in the following way: 

“*They made you feel that it was all okay. You weren’t being judged, you weren’t going to say anything wrong, and they valued your input. So, it just made you want to open up more and be part of it”.*(S14)

The program aimed to enhance a sense of trust in each other. It was designed as a space to help staff self-reflect and heal from the bushfires. The physical spaces that SEED promoted, such as *The Quiet Room* created in one of the hospitals, also aimed to aid in the staff’s healing process. Furnished with comfortable armchairs, and filled with cushions, calming art, candles, colouring books, and other items donated by staff and community members, and overlooking nearby hills (Fieldnotes MUH), the space was welcomed by staff as a “*refuge*”, a place to “*relax, renew*” and “*unwind and prepare for work*” (from the anonymous sticky notes posted in the room). This feeling of emotional safety in a workplace in turn facilitated the practices of creating a sense of connection between staff and caring collectively. 

Several participants described the benefits of creating a safe and healing space as a completely new and different experience in the workplace. For example, “*It just, it life changed me, it propelled me in areas that I can’t explain. I changed, I changed because I felt safe, heard, validated for the first time in my life*” (S16). Other participants described how through this practice they came to realise others shared their experiences and in their work team “*it was okay not to be okay*” (S03). This sense of safety helped staff recognize the impact of the cumulative disasters they were experiencing and permitted them to focus on wellness and healing in their workplace. 

Challenges to this practice were around the workplace and professional culture of healthcare that was noted to often feel unsafe, take a toll on staff, and to cause trauma in the everyday work environment: “*Health is a massive service of experts, and so it’s actually really very difficult to make yourself vulnerable*” (S23). Leadership was specifically referenced in relation to creating safety: “*it’s got to come from the top. You can’t have a safe environment if the most senior person does not facilitate a safe environment*” (S01).

### 3.5. Connecting with Others

SEED was also described as creating connections between staff. During the all-day SEED event, the SEED facilitator emphasized: “*To connect to yourself and connect to each other is what we want to do*”. Opportunities and activities, such as the *Coffee Buddies* and *Wellness Wednesdays*, were introduced in each site so that people could spend some time together and learn about each other. The *Coffee Buddies* activity involved staff being randomly assigned to pairs and going out of the work site to get a coffee together to get to know each other. Coffee vouchers were provided. This is how one staff described this intentional approach to building connections: 

“*The interesting thing about SEED is being able to overtly create those connections on a different level and we do know that when people know each other as human beings, they’re likely to be more compassionate, caring and understanding of each other”.*(S33)

The staff described how important it was to get to know some of their colleagues as a whole person not just the “work person”: “*We got to know each other more because we spent time to check in with each other and we went to Wellness, where we shared things about our life that we didn’t know*” (S03). The benefits of these newly developed connections between staff influenced how they viewed and treated each other and how they felt about work, especially in the aftermath of a disaster and in the midst of a pandemic: 

“*SEED is about enhancing relationships. It’s about making new friends. I’ve made all these new friends right across ISLHD now, which I never would have had the chance to do, and that makes our work easier and our work stronger”.*(S17)

The key challenge of building connections with others was feeling uncomfortable at first and not confident about what was going to happen, especially if the feeling of safety in the workplace was still being established: 

“*So, it was like making us spend time with each other, and for lots of people that was really uncomfortable, you know, it was! [It] was one of the hardest initiatives, because people did not want to go and have coffee with someone they did not know”.*(S03)

### 3.6. Collective Caring

Building a sense of responsibility for the wellbeing of others—a collective caring—was both an intentional approach and an outcome of the SEED program. SEED aimed to create a work culture where colleagues strived to help each other to be well, particularly in the post-disaster recovery period, and they put people before tasks and processes: 

“*In a work environment you can tend to be very professional, but SEED allowed all the barriers to be broken down and that people actually started to really care, as human beings, about one another. Not that they didn’t before, but in a really meaningful and very supportive way”.*(S29)

By having created a safe space and a sense of connection between the workplace, the benefits of collective caring were that staff perceived they were kinder to each other, respected each other and authentically cared for each other: “*I like the idea that there’s a lot more kindness. It’s nice to see someone looking out for somebody else and supporting other staff members, buying them a coffee, recognising that they’re having a tough day, and doing something*” (S12). The anonymous feedback written on the sticky notes in *The Quiet Room*, further demonstrated this new feeling in the workplace: “*Thinking of you all*” and “*This disaster and tragic events has brought us together as a district”,* among others. 

The challenge of collective caring was similar to creating a safe and healing space. The workplace culture of healthcare was identified as a barrier to caring for colleagues because there is an emphasis on being both patient- and task-focused rather than spending time on colleagues: “*I think that we look after people but we’re not good at looking after ourselves at all. We are probably one of the worst groups of people looking after ourselves*” (S09).

### 3.7. Diversifying and Localising Wellness Activities

The SEED program was described as flexible and adaptable to each site to fit the staff needs. Staff could build on the vast array of existing SEED activities, most of which were created at the initial site in response to the bushfires (e.g., *Coffee Buddies*, *Wellness Wednesdays*, yoga, care baskets), and/or create their own. As one staff explained:

“*I think just that it’s got to be site specific, whatever you do. You can’t make this a cookie cutter approach. You’ve really got to find out what the needs of each particular site are because each site works so differently”.*(S20)

Activities were thus being developed and evolved to suit the staff needs over time. Each team/site/ward developed and organised sociomaterial arrangements that fit their needs (e.g., coffee vouchers, *The Quiet Room*, cards, food trucks). SEED promoted and expected a willingness from staff and entire teams to try new ways of working. The key benefit of this practice was that staff created their own localized, place-based approach that suited their context and that they perceived was an effective way to approach wellness: 

“*Some of the things that we do, we have great success because we’re not following a standard framework or model that we’ve been told to do, we’re actually guided by what the staff are thinking and feeling, and then just trying something new”.*(S19)

Challenges to the SEED activities came from staff who did not want to be involved. While this study did not investigate reasons for not participating in SEED, interview participants noted that there could be resentment from staff who did not join in: 

“*There were always people who didn’t receive that kind of support very well. They’re just not people that respond to it, and I know that there’s been similar experiences at [site] where it’s just not their thing, and it’s not everybody’s thing”.*(S21)

### 3.8. Striving for Sustainability

Plans for sustaining wellness activities and culture were discussed from the beginning and a variety of strategies were incorporated to enhance the likelihood of sustainability. One of the ways was having designated time allocations for staff to drive the wellness initiative at their sites: “*having somebody to have designated separate time to continue this work has been really, really important*” (S17). In two sites, one strategy was incorporating a version of the “train the trainer approach” where staff were selected to become the *Wellness Warriors*. The staff participated in a three-day training (two consecutive days and a one-day follow-up one month later) to build their skills in the described practices, build a support network for each other across the local health district and learn how to support other staff (for more details, see Knezevic et al., forthcoming). This is how one staff member described how wellness was sustained in her team, through the *Wellness Warriors* approach:

“*In terms of sustainability, we’ve got a small team that are our Wellness Warriors and they’re the ones that plan our activities and approach us with some of their ideas. There’s about six different things going on in our department that we all contribute to and can be involved in. So, there’s a little planning committee and if someone leaves the department, they will replace that person with another Wellness Warrior to form that little team”.*(S29)

Sustainability of wellness-oriented culture was also closely linked with responsive and compassionate leadership. As one staff described: “*if you’ve got a manager or a director that is supporting it and leadership that supports it [wellness], it’s the first step to ensuring sustainability*” (S29). The low cost and time involved in running SEED was also viewed as helpful factor in sustaining the program: “*So sustainability, I think, is achievable given the commitment of my team and the low cost involved, and there’s not hours and hours’ worth of work involved, and there’s the want*” (S30). 

Nevertheless, various challenges and barriers to sustainability were identified. For example, some staff talked about the overall decrease in a sense of urgency to dedicate time to wellness after the crisis has passed: 

“*I think it was more prioritised when we were in greater crisis. I think we all felt justified. And I think now we’ve done it and … we need to still make more of an effort to keep that going. It’s hard”.*(S03)

In addition, some wards, including the Emergency Department and Coronary Care, found it especially challenging to sustain their wellness activities particularly in face of high staff turnover: 

“*We’re a very high-paced ward. We have lots of turnover. We have very sick patients. Staff do find it very, very challenging to take that time out, even when it’s Where’s Wally on the ward, it’s really hard to draw that attention away from a sick patient”.*(S24)

These challenges required staff to keep re-envisioning when, how, and why to do wellness in high-paced work sites. 

## 4. Discussion

This study explored the participant experiences of the SEED Wellness Program created locally in regional Australia to enhance the resilience and recovery of HCWs post-bushfires and subsequently expanded to respond to staff stress resulting from the COVID-19 pandemic. Participatory narrative inquiry revealed eight interwoven practices that were the core components of the program. Responsive and compassionate leadership was the foundation of all the other practices as implicit and explicit permission for wellness and role modelling from the leaders was essential for launching and maintaining wellness initiatives. SEED was participative and collaborative in nature as it engaged staff at every stage, from initial consultations to assess staff needs at each site to regular opportunities for re-envisioning its structure and components. Other core components of SEED were teaching staff how to hold space for each other, listen to understand, connect with one another, create safe space, and practice collective care. All of these were closely related and built on one another. SEED was also based on the premise that each site needed to develop wellness activities that fit their staff’s needs, resulting in situated and tailored wellness practice approaches rather than implementing a one-size fits all intervention. Finally, sustainability was frequently discussed, and actions were taken to promote wellness-orientation at an organisational level and to ensure that each ward had its own mechanisms in place to ensure the continuation of wellness activities beyond the crisis. Although the study’s aim was not to conduct evaluation of SEED, the satisfaction of staff participating in the study was almost universal. 

The findings emphasise the importance of creating connections between staff for mutual support. This practice is consistent with Hobfoll’s [13] principle of connectedness in post-disaster resilience and recovery interventions. Social connections and group involvement have been positively linked to mental health outcomes following disaster [49,50]. Conversely, bushfire survivors who had fewer social connections were at higher risk for depression and PTSD [49]. The COVID-19 outbreak, and the resulting social distancing and lockdown measures have substantially limited people’s ability to connect and provide support to each other in a work environment already known for low levels of social support [5]. SEED promoted and facilitated workplace group activities that focused on engaging with people at a personal level rather than on workplace tasks. SEED views the work team as a community requiring a community-level intervention, an intervention that has been demonstrated as effective in enhancing recovery [32]. Further, SEED gave participants skills to connect with and support others, which in similar disasters has given people confidence to step in and assist in the recovery effort [51]. 

A key SEED practice of co-design allowed for the broad engagement of all team members including administration, security, nurses, and leaders, rather than focusing on any single group. This practice recognised the commonality of the workplace experience and emphasised that healthcare delivery requires all roles to work together. It is the system and work community that faces uncertainty and must constantly adapt, not just a single profession [22]. The co-design at each site meant that work teams felt ownership of SEED activities and that these activities incorporated the team’s specific work environment and work demands so were place-based and grounded in local knowledge, needs and resources. 

### Implications for Policy and Practice

The pandemic has in some way given HCWs and hospitals permission to recognise, report and respond to workplace stresses that were well documented prior to 2020 [4]. Many studies have documented the impact of the pandemic on HCWs and called for interventions that address emotional exhaustion, burnout, and stress [23,24,25,26]. However, few programs have emerged to date. The ISLHD initiated and supported SEED because the organisation’s leadership recognised the staff need for mental health support and quickly provided the staff and resources able to deliver this support. Ability to maintain core functions and activities has been identified as a key factor in healthcare resilience during the pandemic [22]. However, both bottom-up and top-down commitment is required to sustain wellness-oriented practices. To implement and sustain wellness programs leaders must facilitate staff participation through both role-modelling wellness and supporting staff engagement in activities. 

Healthcare organisations are concerned with recruiting, retaining, and supporting staff who can provide quality care to patients [28]. Multiple studies identify the need for staff support programs [23,24,25,26,27,28]. Workforce strategies need to be embedded in and delivered via policy initiatives that ensure adequate resourcing to facilitate staff participation in wellness programs. In a resource stretched environment such as healthcare, it may be difficult to see staff wellness programs as a legitimate expenditure when patient care is the priority. However, improved HCWs wellbeing has a flow-on effect to better patient outcomes [52], and the recognition of this relationship is vital. 

Workplace health promotion has tended to focus on an individual’s health status while at work rather than seeing health status as affected by work. The eight SEED practices outlined in this article can provide guidance to healthcare organisations and policy makers about one way to support staff that moves away from placing the responsibility for managing workplace stress on to individuals and their management of lifestyle factors [29]. 

## 5. Strengths and Limitations of the Study

Studies of COVID-19 impacts on healthcare staff unanimously call for wellness interventions to be implemented. This study is one of the few to date that have investigated participant experiences of a locally designed and implemented wellness program. Further, the practice theory approach has enabled detailed description of the key SEED practices that can be replicated for further research. The study, however, is limited to one local health district and focused on the unique features of that district. In addition, the study focused only on HCW who participated in the SEED Wellness Program, and it did not include those who did not participate. It is also important to note that due to the increased workloads of HCW during the COVID-19 pandemic and thus a limited capacity to participate in research, the recruitment of HCW proved difficult in the later stages of the project. Another potential limitation is that the participants were largely female and White therefore the results may not be reflective of the wellness needs of men or people from other cultural groups. Finally, the study did not formally evaluate the outcomes of the SEED program. Considering these limitations, plans for conducting a stepped wedge cluster randomised trial to assess SEED outcomes are currently under way. 

## 6. Conclusions

While workplace wellness is not a new concept, it has gained a new meaning and significance in the healthcare workforce in face of the COVID-19 pandemic. Some healthcare workers, like those interviewed for this study, had to carry the burden of responding to multiples crises—the impact of the pandemic compounded by prior natural disasters. Although wellness is often perceived as an individual responsibility that should be addressed outside the workplace, this study demonstrated that workplace wellness interventions may be needed to not only prevent staff burnout and turnover but also to lead organisational culture change. This study detailed the practices that made up a successful hospital wellness intervention including connecting with others and localising wellness activities. In healthcare organisations, focusing on staff wellness can ultimately lead to better care and outcomes for their patients.

## Figures and Tables

**Table 1 ijerph-19-13204-t001:** Participant demographics.

Gender (*n*)	Ethnicity (*n*)	Mean Age in Years (Range)	Work Role (*n*)	Mean Number of Years in Current Role (Range)
Female 29 Male 4	Aboriginal Australian 3 Caucasian 26 Other 4	49.9 (32–65)	Nurse * 9 Nurse Educator 5 Manager/Executive 10 Health and Security 2 Allied Health 2 Administration 3 Project Manager 2	7.5 (1–37)

Note. * Nurse includes all patient facing roles (Registered Nurse, Enrolled Nurse, Clinical Nurse Specialist).

## Data Availability

The data presented in this study are available on request from the corresponding author. The data are not publicly available due to containing potentially identifying information as the study was conducted in small, regional hospitals in one local health district.

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
