# Peer review of "A Narrative Inquiry into the Practices of Healthcare Workers’ Wellness Program: The SEED Experience in New South Wales, Australia"

_ijerph, 2022, doi:10.3390/ijerph192013204_

Round 1

Reviewer 1 Report

Thank you for the opportunity to review this interesting research manuscript. Overall, the manuscript is well-written and provides some very useful insights into the somewhat neglected area of healthcare staff wellbeing. An original method of co-design has been employed, and the findings seem all the richer for that. I agree with the authors' view that there is little by way of intervention studies in this area. To that end, this paper will make an interesting and important contribution.

I encourage the authors to consider the following matters to strengthen the contribution this paper can make:

- conceptually, some further clarifications of the various perspectives used to inform this study could be helpful. The abstract refers to participatory action methodology, whereas PNI is then referred to in the methods. Similarly, a seemingly generic sounding 'Practice Theory' is introduced in the methods section (an approach admittedly I have never heard of). It could be helpful to consider a section on methodology, before methods, and more fully elaborate these approaches so that how they inform the study can be more clearly understood. This could then inform the discussion and conclusions more robustly?

- while the aim is to describe practices, it would be helpful to link these with other disaster principles. The Hobfoll et al five principles of mass disaster intervention resonate strongly with these practices identified in this study, and a more recent paper by Bryant et al has applied these to COVID. This paper could link well by demonstrating how these principles of intervention can be translated into practices. At the moment, the study sits outside of some of this foundational disaster literature and could be strengthened by it.

- in relation to the data collection, some detail as to who the focus group participants were would be helpful - it seems as if 12 people in total must have participated in them? In the limitations, these issues of recruitment could be addressed.

- the data analysis refers to 23 transcripts, yet 21 interviews and 3 focus groups are referred to. Should this be 24 transcripts?

- could further information be provided as to how the specific questions asked of each transcript translated into the collaborative thematic analysis? Were these separate or inductive processes? Given the findings then report on 8 specific practices, were these the culmination of both these processes of analysis, or more the latter?

- curiously, the results and practices outlined seem devoid of a specific positioning in a post-disaster and COVID context. Did the data provide insight into how people were seeing these practices as distinctive because of the disaster context? At times, the findings read as if they could be the generic findings from any staff wellbeing program, so some deeper characterisation and differentiation of context would be invaluable if possible.

- on a minor note, within the manuscript, the formatting of quotations has meant they are indistinct from the body text. I am assuming on copy-editing this matter would be addressed.

- similarly on a minor note, there seems to be an overly-extensive reference list. Some editing back would be appropriate.

Reviewer 2 Report

The manuscript “A Narrative Inquiry into the Practices of Healthcare Workers 1 Wellness Program: The SEED Experience in New South Wales, Australia” has significant in the context of

SEED used many times without any abbreviation.

Please describe “practices of a hospital staff wellness program named SEED” in the beginning of introduction and in abstract.

NSW abbreviate it in abstract.

Introduction

Overall, the Introduction section is lengthy and repetitive. The introduction needs to be more concise and coherent.

PTSD, ISLHD abbreviate.

Methods

Aboriginal Community Controlled Health Organisation (for details, see upcoming publications); what do you mean here?

Provide a table for participants detail (characteristics).

Remove the sentence

“How did SEED begin for you?”, “What does SEED look like in your workplace?”, “How would you describe SEED’s philosophy? / What is SEED about?” in line 157

The questions asked of each transcript were: 180 • How is SEED described? What are its practices?   What do people say about their own or others' wellness or un-wellness and ways that SEED influenced that? How are the absence or presence of resources described?  What relationships between people are discussed? How do people do things differently because of SEED (line 181 to 185).  

Provide a table on the Interview guide and Focus Group Discussion guide.

Mention the method, source and investigator triangulation.

In the “Data Analysis” section add about data management.

Results

Provide a table on the Coding tree – themes and codes  

You have a discussion section Why do you give references in results? This is your study findings.

The results must be written in the participant's language, and what the participants shared/viewed/perceived. However, the whole results section is described at an interpretive level.

Discussion

Add one paragraph on implications for policy and practices

Instead of limitations write Methodological considerations and present both strengths and limitations of the study including the credibility. 

Round 2

Reviewer 2 Report

Thanks for addressing all the comments.